# REGULARIZED OPTIMAL TRANSPORT FOR TEMPORAL TRAJECTORY ANALYSIS IN SINGLE-CELL DATA

## ABSTRACT

The temporal relationship between different cellular states and lineages is only partially understood and has major significance for cell differentiation and cancer progression. However, two pain points persist and limit learning-based solutions: (*a*) lack of real datasets and standardized benchmark for early cell developments; (*b*) the complicated transcriptional data fail classic temporal analyses. We integrate `Mouse-RGC`, a large-scale mouse retinal ganglion cell dataset with annotations for 9 time stages and $30,000$ gene expressions. Existing approaches show a limited generalization of our datasets. To tackle the modeling bottleneck, we then translate this fundamental biology problem into a machine learning formulation, *i.e.*, *temporal trajectory analysis*. An innovative regularized optimal transport algorithm, `TAROT`, is proposed to fill in the research gap, consisting of (1) customized masked autoencoder to extract high-quality cell representations; (2) cost function regularization through biology priors for distribution transports; (3) continuous temporal trajectory optimization based on discrete matched time stages. Extensive empirical investigations demonstrate that our framework produces superior cell lineages and pseudotime, compared to existing approaches on `Mouse-RGC` and another two public benchmarks. Moreover, `TAROT` is capable of identifying biologically meaningful gene sets along with the developmental trajectory, and its simulated gene knockout results echo the findings in physical wet lab validation. Codes are provided in the supplement.

## 1 INTRODUCTION

Since first introduced in 2009, large-scale single-cell RNA sequencing (scRNA-seq) has presented enormous opportunities for researchers in various research fields (Patel et al., 2014; Satija et al., 2015; Tirosh et al., 2016). It helps reveal detailed information on transcriptional patterns in different cell and tissue types as well as disease models (Elmentaite et al., 2022; Jagadeesh et al., 2022). Equipped with scRNA-seq, we are able to discover significant heterogeneities that would never be found with bulk analysis within the cell population, which contributes to understanding biology questions with higher cellular resolution. The fast-advancing technology and increased recognition of different cell subtypes also naturally lead us to ask: ① *How and when are the cell subtypes established?* ② *Could we predict the developmental trajectory of each cell and predict the cell "fate" based on current status?* ③ *And could we find the key regulator that controls this type of establishment?* Answers to those questions are important

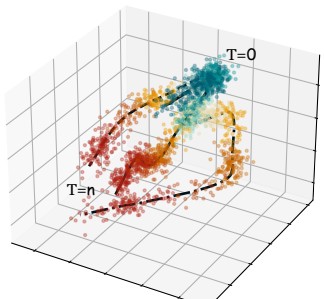

Figure 1: Demo Cell Temporal Trajectories from Time $0 \to n$. Different colors indicate cells from different time stages.

for cell differentiation research in developmental biology (Rizvi et al., 2017; Han et al., 2018; Gulati et al., 2020) and can provide promising pipelines to demystify the cellular response during disease progression (Zhang et al., 2021; Jia et al., 2022). In the past decade, although a great amount of effort (Trapnell et al., 2014; Qiu et al., 2017; Ji & Ji, 2016; Street et al., 2018; Cao et al., 2019) has been put into developing trajectory inference methods using single-cell sequencing data, it remains extremely challenging. This is because, with current technologies, we can not trace the same population of cells over developmental time. It only allows us to collect the transcriptional information of cells for a specific time point as a "snapshot", and then a sophisticated computational model-

ing (Saelens et al., 2019; Van den Berge et al., 2020) is required to construct cell trajectories over multiple "snapshots", as demonstrated in Figure 1. Existing algorithms reach good performance on simulated datasets (Klein et al., 2023) but are still unsatisfactory on realistic benchmarks.

To enhance the capabilities of learning-based algorithms, we generate `Mouse-RGC`, which is a large-scale integrated mouse retinal ganglion cell dataset. It contains $30,000$ gene expressions from 9 time stages of early cell development. However, naively plugging previous approaches (Street et al., 2018; Klein et al., 2023) fail to generalize well on our benchmark, implying their shortage in handling real cases with much higher data complexity. To develop effective solutions, we recast the biology challenge into a machine learning problem, *i.e.*, *temporal trajectory analysis*, aiming to transport cells across time stages. In detail, our proposed `TAROT` first learns superior cell representations through a tailored masked autoencoder. Then, it performs a regularized optimal transport (OT) to produce mappings between every two-time stages. During the matching, we consider the biological priors of gene expression from both developmental and functional perspectives. Note that directly applying OT will result in inferior results due to neglecting the intrinsic structures in this biology problem. Last, continuous temporal trajectories (*i.e.*, cell pseudotime) are optimized and generated by fitting ordered discrete time stages. Our contributions are summarized below:

* We integrate a larger-scale scRNA-seq dataset, *i.e.*, `Mouse-RGC`, with $30,000$ mouse neuron cells across annotations of 9 early developmental time stages. It provides a standardized and challenging benchmark for further research in machine learning (ML) and single-cell transcriptomics.

* We recast the analyses of cell developmental differentiation as an ML problem of inferring temporal trajectories. Our proposed `TAROT` consists of an improved design of cell representation extractor and regularized OT with biology priors, delivering substantially enhanced cell lineages.

* Based on discrete inferred lineages, we introduce B-Splines optimization to produce continuous cell pseudotime estimations with superior quality.

* Extensive experiments validate the effectiveness of our proposals on `Mouse-RGC` and two public datasets. For example, `TAROT` achieves $\{3.10\% \sim 65.03\%, 13.70\% \sim 35.08\%, 6.16\% \sim 27.49\%, 20.82\% \sim 44.28\%\}$ performance improvements on `Mouse-RGC` and `Mouse-MCC` datasets over previous approaches.

* Moreover, `TAROT` can locate crucial gene sets that are biologically meaningful for each temporal trajectory. Removing these genes significantly reshapes the simulated cell differentiation, echoed with the wet lab studies on the `Mouse-iPE` dataset.

## 2  RELATED WORKS

**Optimal Transport (OT).**  OT (Villani et al., 2009; Peyré et al., 2019) serves as a powerful tool for comparing two measures in a Lagrangian framework. It has played a beneficial role in widespread applications in statistics (Munk & Czado, 1998; Evans & Matsen, 2012; Sommerfeld & Munk, 2018; Goldfeld et al., 2022) and machine learning (Schmitz et al., 2018; Kolouri et al., 2018) domains. OT can also be used to define metrics such as the Wasserstein distance (Arjovsky et al., 2017; Liu et al., 2019), which has gained tremendous popularity in the training of generative adversary networks (Deshpande et al., 2019; Adler & Lunz, 2018; Petzka et al., 2017; Deshpande et al., 2018; Yang et al., 2018; Baumgartner et al., 2018; Wu et al., 2019), transfer learning (Shen et al., 2018; Lee et al., 2019), and contrastive representation learning (Chen et al., 2021). There also are several preliminary studies to model the cellular dynamics network (Tong et al., 2020) and cell developmental trajectories (Schiebinger et al., 2019; Klein et al., 2023) through OT.

**Representation Learning in Single-Cell Genomics.**  Extracting powerful cell representations is one of the ultra goals for single-cell genomics. It has been investigated for a long history, and various solutions are delivered ranging from classic optimization algorithm (Li et al., 2017; Satija et al., 2015; Zhao et al., 2022; Stuart et al., 2019) to modern deep learning-based approaches (Yang et al., 2022; Hao et al., 2023; Cui et al., 2022; Geuenich et al., 2023; Zhao et al., 2023). For instance, Yang et al. (2022) advocates the bi-directional transformer to learn a robust single-cell representation. To further improve the quality of learned cell representation, more recent studies leverage a variety of advanced pre-training designs, including generative (Shen et al., 2023; Cui et al., 2023), mask language modeling (Hao et al., 2023), multi-task learning (Cui et al., 2022), self-supervised active learning (Geuenich et al., 2023), and contrastive learning (Zhao et al., 2023) objectives.

**Lineage and Pseudotime Inference.** The increasing availability of scRNA-seq data allows researchers to reconstruct the trajectories of cells during a dynamic process. The relationships between different cellular states and lineages are extremely important for studies on embryonic development (Griffiths et al., 2018; Cang et al., 2021; Mittnenzweig et al., 2021; Kim et al., 2023), cell differentiation (Rizvi et al., 2017; Han et al., 2018; Gulati et al., 2020), cancer progression (Zhang et al., 2021; Jia et al., 2022) and cell fate diversification (Buchholz et al., 2016; Koenig et al., 2022). In the past few years, numerous trajectory inference pipelines have been established, which can be roughly divided into two major categories based on the algorithm they used. The first and perhaps the most commonly used one is minimum spanning tree (MST) based approaches. Monocle and Monocle-2, which are the early used methods, both infer the developmental trajectory of one single cell level and assign the pseudotime of each cell (Trapnell et al., 2014; Qiu et al., 2017). Later, Tools for Single Cell Analysis (TSCAN) (Ji & Ji, 2016) and Slingshot (Street et al., 2018) run the MST algorithm on clusters to construct the cluster-based MST. Then, they orthogonally project each cell onto the paths of the MST to get the pseudotime. Notably, Slingshot utilized a principal curves algorithm to calculate smooth curves from MST, which gives better visualization. The second category is the graph-based trajectory inference method, which uses multiple algorithms to construct trajectories among cells. One famous and widely used tool, Monocle3 (Cao et al., 2019), generates the trajectory using a principal graph algorithm. Then, it calculates the shortest Euclidean distance of each cell from the root node to assign the pseudotime. However, the self-selected root node required some prior knowledge about the cell identity. Diffusion pseudotime (DPT) (Haghverdi et al., 2016) and URD (Farrell et al., 2018) uses a $k$-nearest-neighbor algorithm to construct the temporal trajectory of the cells in gene expression space.

**Single-Cell Transcriptomics.** The heterogeneity analysis is the core reason for performing single-cell sequencing studies. It assesses the transcriptional similarities and differences within the cell populations and helps reveal a higher cellular resolution among cells (Haque et al., 2017; Satija et al., 2015; Tirosh et al., 2016). Using scRNA-seq (Patel et al., 2014), researchers are able to define detailed heterogeneity of immune cells (Shalek et al., 2013; Mahata et al., 2014; Stubbington et al., 2017), cancer cells (Wu et al., 2021; Fan et al., 2020), embryonic stem cells (Jaitin et al., 2014; Klein et al., 2015) *etc*. In the meantime, transcriptional assessments with single-cell sequencing technology also identify rare cell populations that would never been detected using bulk analysis (Miyamoto et al., 2015; Zeisel et al., 2015; Tirosh et al., 2016). Equally important, the gene co-expression patterns that scRNA-seq reveals allow us to define gene modules and point out the underlying mechanism of gene expression regulations (Wagner et al., 2016).

## 3 DATASET AND MACHINE LEARNING FORMULATION

### 3.1 MOUSE-RGC: A LARGE-SCALE DATASET OF RETINAL GANGLION CELLS FROM MOUSE

In this section, we introduce all three datasets that are adopted to evaluate TAROT's effectiveness. As for public datasets, we consider a mouse cerebral cortex cell benchmark (Di Bella et al., 2021), *i.e.*, Mouse-CCC, and a mouse induced Erythroid Progenitor (iEP)-derived cell benchmark (Capellera-Garcia et al., 2016), *i.e.*, Mouse-iEP, which contains {66443, 1947} cells across {11, 2} time stages, respectively. The detailed information about our Mouse-RGC is presented below.

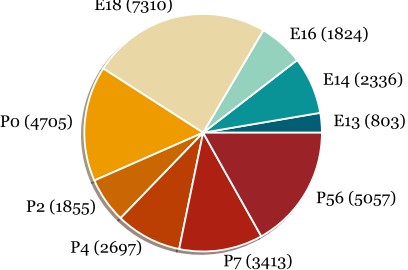

Figure 2: The sample distribution of our Mouse-RGC (30K cells) dataset based on developmental time stages. For example, "E18 (7310)" indicates that there are 7,310 cell samples in time stage E18.

**Data Collection.** To generate the Mouse-RGC dataset, we extract 30K mouse neuron cells from previously published datasets (ref 1,2) and newly formed data. The developmental time stages of {E13, E14, E16, E18, P0, P2, P4, P7, P56}, as shown in Figure 4. Then, the corresponding gene expressions are measured by the RNA sequencing technique as previously defined. Single-cell libraries were prepared using the single-cell gene expression 3′ kit on the Chromium platform (10X Genomics, Pleasanton, CA) following the manufacturer's protocol. To be specific, single cells were partitioned into Gel beads in EMulsion (GEMs) in the 10X Chromium instrument followed by cell lysis and barcoded reverse transcription of RNA, amplification, enzymatic fragmentation, 5′ adaptor attachment, and sample indexing. On average, approximately $8,000 \sim 12,000$

single cells were loaded on each channel, and approximately $3,000 \sim 7,000$ cells were recovered. Libraries were sequenced on the Illumina HiSeq $2,500$ platforms.

**Preprocess and Properties.** After we collected the raw signals, the following single-cell sequencing data processing was done using the Seurat package (Hao et al., 2021). Sample quality control was performed on each sample individually. For each sample, doublets were removed using DoubletFinder (McGinnis et al., 2019). We retained cells that expressed at least $1,500$ genes and less than $11,000$ genes. Meanwhile, we removed cells that have more than $5\%$ mitochondrial genes and genes expressed in fewer than 10 cells. The resulting $n$ cells $\times$ $g$ genes matrix of UMI counts were subject to downstream analysis. The UMI-based gene expression matrix was normalized using sctransform (Hafemeister & Satija, 2019). After that, the batch correction was done with canonical correlation analysis (Hotelling, 1992; Anderson et al., 1958), using the top $4,000$ anchor genes.

**Clustering.** In this research, we are interested in the evolution of different cell types of mouse neurons. Therefore, we built a nearest-neighbor graph to cluster cells based on their transcriptional similarity. Specifically, the number of nearest neighbors was chosen to be 50, according to the rich experiences of biology scientists. The edges were weighted based on the Jaccard overlap metric, and graph clustering was performed using the Louvain algorithm (Blondel et al., 2008). In the end, as demonstrated in Figure 3 (*Left*), the cell clusters were then projected onto a nonlinear 2D space using the Uniform Manifold Approximation and Projection (UMAP) algorithm (McInnes et al., 2018). For temporal trajectory analysis, we further decomposed the cell clusters by their time stage labels like Figure 3 (*Right*). Our goal is to demystify the neuron evolution path across these 9 time stages.

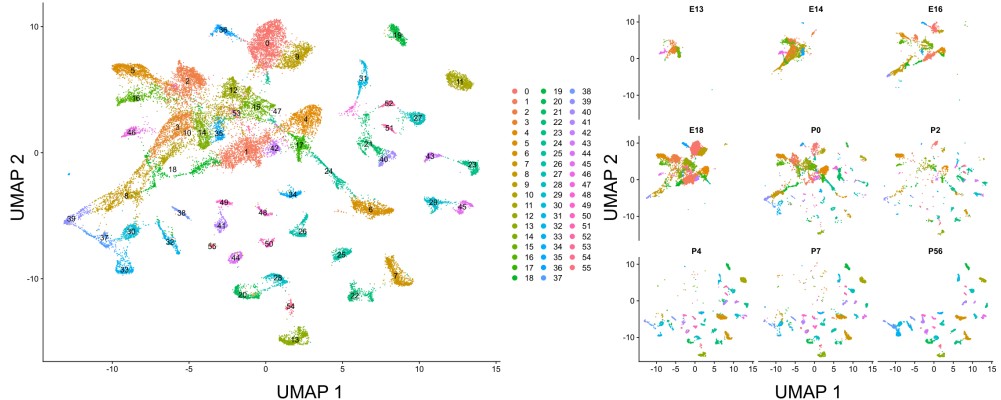

Figure 3: *(Left)*: Clustering `Mouse-RGC` to 56 kinds of cell types and projecting them into a 2D space via UMAP; *(Right)*: Decomposing the clustering results by their time stage labels. Zoom-in for better reliability.

## 3.2 ML Formulation - Matching Sample Distribution across Temporal Stages

Understanding the development of stem cells into fully differentiated cells requires accurate cell lineage and pseudotime. Thus, the fundamental biology problem here is *how to model and infer evaluation trajectory of neurons?* This paper translates and recasts it as a machine learning (ML) problem, aiming to *match cell distributions across temporal stages*.

**Notations.** Let $\{r_i\}_{i=1}^n$ denote the raw cell expressions and $\{c_i\}_{i=1}^n$ are extracted cell representations, where $n$ is the total number of cells and $r_i \in \mathbb{R}^{1 \times g}$. For each cell representation $c_i \in \{c_1, \cdots, c_n\}$, it has the labels of time stage and cluster, obtained from the data preprocessing. Therefore, the total $n$ cells can be divided into $k$ groups, *i.e.*, $\{\mathcal{C}_1, \cdots, \mathcal{C}_k\}$ where $\sum_{i=1}^k |\mathcal{C}_i| = \sum_{i=1}^k n_i = n$, $\mathcal{C}_i = \{c_1^{(i)}, \cdots, c_{n_i}^{(i)}\}$, $|\mathcal{C}_i| = n_i$ is the number of cells in cluster $\mathcal{C}_i$. Considering temporal information like time stages $\{t\}_{t=1}^s$, we use $\mathcal{G}^{(t)} = \{\mathcal{C}_1^{(t)}, \cdots, \mathcal{C}_{k_t}^{(t)}\}$ to represent all cells in the time stage $t$, where $s$ is the total number of time stages and $k_t$ denotes the number of clusters at time step $t$. Our goal is to establish a mapping from $\mathcal{G}^t \to \mathcal{G}^{(t+1)}$, which shares certain similarity to the trajectory analysis problem (Helland-Hansen & Hampson, 2009).

**Problem Definition.** *Given the cluster set $\mathcal{G}^{(t)} = \{\mathcal{C}_1^{(t)}, \cdots, \mathcal{C}_{k_t}^{(t)}\}$ as each time stage $t \in \{1, \cdots, s\}$, we aim to (1) infer temporal trajectories like $\mathcal{G}^{(1)} \to \mathcal{G}^{(2)} \to \cdots \to \mathcal{G}^{(s)}$, based on their gene expression; (2) estimate continuous pseudotime for each sample.*

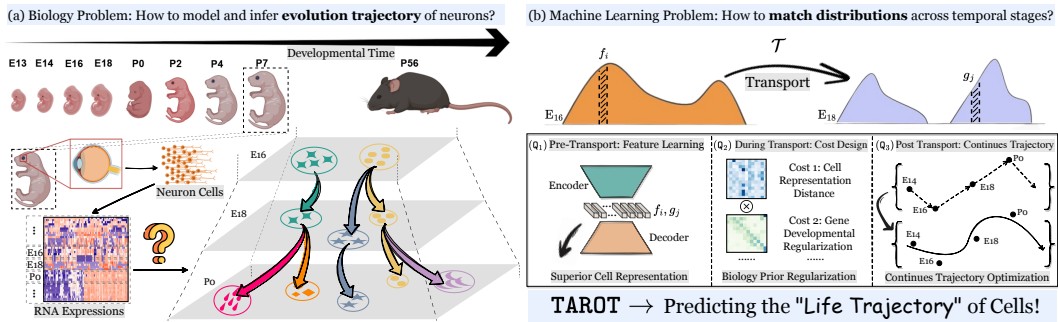

Figure 4: (a) *Biology Problem.* We aim to model and infer the evolution trajectory of neurons. Specifically, the neuron cells are extracted from the developmental time stages {E13, E14, E16, P0, P2, P4, P7, P56} of mouses. Then, RNA sequencing is performed to collect its expression data. (b) *Machine Learning Problem.* We translate the biology problem to an ML problem – matching sample distributions across multiple temporal stages. This challenging transport problem can be further decomposed into three sub-questions, *i.e.*, $\mathbb{Q}_1$, $\mathbb{Q}_2$, and $\mathbb{Q}_3$. To tackle these research questions, our proposed TAROT introduces superior cell representations, regularized optimal transport via biology priors, and continuous trajectory optimization, respectively.

**An Ideal Solution.** To infer the temporal trajectory, it requires answering three key questions ($\mathbb{Q}_1$, $\mathbb{Q}_2$, and $\mathbb{Q}_3$) as summarized in Figure 4 (b):

① *Before the Transportation.* It needs to extract high-quality cell representations $\{\mathbf{c}_i\}_{i=1}^n$ from the gene expressions $\{\mathbf{r}_i\}_{i=1}^n$. Both low-dimensional projection methods like PCA (Bro & Smilde, 2014) and UMAP (McInnes et al., 2018), and deep neural networks (Yang et al., 2022; Shen et al., 2023; Cui et al., 2023) can serve as the desired feature extractor.

② *During the Transportation.* It focuses on computing the mapping function $\mathcal{T}_{t,t+1} : \mathcal{G}^{(t)} \to \mathcal{G}^{(t+1)}$, $t \in \{1, \cdots, s-1\}$, given all cell information from the current and history time stages. Each mapping between two-time stages is a bipartite graph and can be derived from distribution matching problems (Gretton et al., 2012) through the Hungarian algorithm or Optimal Transport, *etc*. The crucial challenge here is the design of cost functions for transportation. Naively plugging in distance measurements based on ML intuitions leads to inferior results (Klein et al., 2023), which demands appropriate cost designs to integrate *biology priors* of the neuron developments.

③ *After the Transportation.* Global lineages are deduced according to the pair-wised mapping $\{\mathcal{T}_{t,t+1}\}_{t=1}^{s-1}$. However, it only contains a discrete order of different time stages, which is an irregularly sampled time series due to the constraints of cell data collection. Since cell differentiation occurs continuously, we need to calculate a continuous cell pseudotime based on its inferred lineage. It can be addressed by interpolation approaches (Shukla & Marlin, 2020) like Splines.

## 4 METHODOLOGY

**Overview of TAROT.** The overall procedures of TAROT are described in Figure 4. Our proposal tackles the aforementioned ML problem by answering the three key questions. Before transport, we introduce a customized Masked Autoencoder (MAE) transformer to learn adequate cell representation. During transport, we integrate important biology priors into the design of cost functions and leverage them to enable a regularized optimal transport. After transport, continuous trajectories will be produced by performing B-Splines fitting optimization to inferred cell lineages.

### 4.1 TAROT: CELL LINEAGE INFERENCE VIA DISTRIBUTION MATCHING

**Regularized Optimal Transport for Matching.** Optimal transport (OT) distance is a popular option for comparing two distributions. We consider the discrete situation in our case. For two time steps $t_1$ and $t_2$, there are two sets of features $\{\boldsymbol{f}_i\}_{i=1}^{\mathrm{M}}$ and $\{\boldsymbol{g}_j\}_{j=1}^{\mathrm{N}}$. Since we focus on a cluster-level mapping, then $\mathrm{M} = k_{t_1}$ and $\mathrm{N} = k_{t_2}$ are the number of clusters in stage $t_1$ and $t_2$ respectively. $\boldsymbol{f}_i$ and $\boldsymbol{g}_j$ are averaged cell representations for each cluster. Note that it is straightforward to extend to cell-level mapping by adopting cell-specific representations. Our discrete distributions can be formulated as $\boldsymbol{u} = \sum_{i=1}^{\mathrm{M}} u_i \delta_{\boldsymbol{f}_i}$ and $\boldsymbol{v} = \sum_{j=1}^{\mathrm{N}} v_j \delta_{\boldsymbol{g}_j}$, where $\boldsymbol{u}$ and $\boldsymbol{v}$ are the discrete probability vectors that

sum to 1, and $\delta_{\boldsymbol{f}}$ (or $\delta_{\boldsymbol{g}}$) is a Dirac $\delta$ function placed at support point $\boldsymbol{f}$ (or $\boldsymbol{g}$) in the embedding space. Then, the total cost of transportation is depicted as $<\mathcal{T}, \mathcal{D}> = \sum_{i=1}^{M} \sum_{j=1}^{N} \mathcal{T}_{i,j} \mathcal{D}_{i,j}$.

The matrix $\mathcal{D}$ is a cost matrix, where each element denotes the cost between feature $\boldsymbol{f}_i$ and $\boldsymbol{g}_j$, like $\mathcal{D}_{i,j} = 1 - \text{sim}(\boldsymbol{f}_i, \boldsymbol{g}_j)$ and $\text{sim}(\cdot, \cdot)$ is a similarity measuring function. The $\mathcal{T}$ is the transport matrix that describes the mapping from $\{\boldsymbol{f}_i\}_{i=1}^{M}$ to $\{\boldsymbol{g}_j\}_{j=1}^{N}$. To learn the transport plan $\mathcal{T}$, it will minimize the total cost as follows:

$$\mathcal{F}_{\text{OT}}(\boldsymbol{u}, \boldsymbol{v}|\mathcal{D}) = \min_{\mathcal{T}} \langle \mathcal{T}, \mathcal{D} \rangle \tag{1}$$

$$\text{s.t. } \mathcal{D} \times \mathbf{1}_N = \boldsymbol{u}, \ \mathcal{D}^T \times \mathbf{1}_M = \boldsymbol{v}, \ \mathcal{D} \in \mathbb{R}_+^{M \times N}. \tag{2}$$

However, this formulation has a super-cubic complexity in the size of $\boldsymbol{u}$ and $\boldsymbol{v}$, which prevents adapting OT in large-scale scenarios. Sinkhorn algorithm Cuturi (2013) is applied to speed up the computation via an entropy regularization, *i.e.*, $\mathcal{F}_{\text{OT}}(\boldsymbol{u}, \boldsymbol{v}|\mathcal{D}) = \min_{\mathcal{T}} < \mathcal{T}, \mathcal{D} > -\lambda \mathcal{E}(\mathcal{D})$, where $\mathcal{E}(\cdot)$ is the entropy function and $\lambda \geq 0$ is a hyper-parameter. The optimization of $\mathcal{F}_{\text{OT}}$ constitutes the base framework of TAROT, and more innovative designs are described as follows.

**Cell Representations via Masked Autoencoder Transformers (MAE).** TAROT tailors an MAE transformer to extract superior cell representations from the gene expressions $\{\mathbf{r}_i\}_{i=1}^{n}$. Figure 5 illustrates the MAE procedure: the raw signals are first masked and fed into the MAE encoder; then, masked

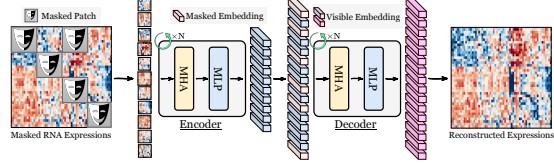

Figure 5: The overall procedure of MAE in TAROT.

embeddings are incorporated to align with the full input dimensions; finally, the decoder reconstructs the original input data and computes the MSE training objective. In the inference phase, TAROT adopts the MAE encoder to generate cell representations.

**Biology Priors Regularize Cost Function.** Another critical component is the cost function design. TAROT takes two essential biological factors into account from neuron developmental and gene expression perspectives. ① (*Developmental*) *The natural cell differentiation never look back.* In other words, clusters in $\mathcal{G}^{(t)}$ can not be mapped to their ancestor clusters from history trajectories $\{\mathcal{T}_{1,2}, \cdots, \mathcal{T}_{t-1,t}\}$. Specifically, an extra cost penalty $\mathcal{D}_{i,j}^{\text{dev}}$ will be applied if the cluster $j$ from $\mathcal{G}^{(t+1)}$ at time $t+1$ is an ancestor of the cluster $i$ from $\mathcal{G}^{(t)}$ at time $t$. ② (*Gene expression*) *The expressions of developmental-related genes satisfy particular patterns.* A specific group of genes tends to have monotonically increased expressions along with the cell differentiation trajectories. If a mapping $\mathcal{T}_{t,t+1}$ does not meet this prior, an additional cost penalty $\mathcal{D}_{i,j}^{\text{fuc}}$ will be introduced. Considering these two biology regulations (①+②), the final cost function is depicted as $\tilde{\mathcal{D}} = (\mathcal{D}^{\text{dev}} + \mathcal{D}^{\text{fuc}}) \odot \mathcal{D}$, where $\odot$ denotes the element-wise product and $\mathcal{D} = 1 - \text{corr}(\mathcal{G}^{(t)}, \mathcal{G}^{(t+1)})$ is a vanilla cost based on the correlation (*i.e.*, $\text{corr}(\cdot, \cdot)$) of cell representations.

## 4.2 TAROT: PSEUDOTIME CALCULATION VIA CONTINUOUS TRAJECTORY OPTIMIZATION

Cell lineages with discrete time orders are produced from the first stage of TAROT. To estimate the cell pseudotime, TAROT executes fitting optimization for continuous temporal trajectories.

**Continuous Trajectory via B-Splines.** A K-degree B-Spline is defined as $\mathcal{C}(u) = \sum_{i=0}^{I} \mathcal{N}_{i,k}(u) \cdot p_i$, where $\{\mathcal{N}_{i,k}(\cdot)\}_{i=0,k=0}^{i=I-1,k=K}$ are bases and the I is the number of control points $\{p_i\}$.

More details about the bases of B-Splines are provided in Appendix A1. In TAROT, for each lineage, we insert J learnable control points $\{p_i^{(j)}\}|_{j=1}^{J}$ between two fixed control points $p_i$ and $p_{i+1}$. TAROT treats the averaged cell presentation of each cluster as the fixed control point. And the continuous trajectory optimization (Figure 4 - *Right*) is described as $\min_{\{p_i^{(j)}\}_{i=0,j=1}^{i=I-1,j=J}} \sum_{k=0}^{n'} \|\mathbf{c}_k - \mathbb{P}(\mathbf{c}_k, \mathcal{C}(u))\|^2$, where $\mathbb{P}(\mathbf{c}_k, \mathcal{C}(u))$ is the projection of cell representation $\mathbf{c}_k$ on $\mathcal{C}(u)$, and $n'$ is the total number of cells on the lineage. Then, the pseudotime $u(\mathbf{c}_k)$ is derived as $\text{argmin}_u \|\mathbf{c}_k - \mathcal{C}(u)\|^2, u \in [0, 1]$.

## 5 EXPERIMENTS

### 5.1 IMPLEMENTATION DETAILS

**Evaluation Metrics.** We introduce five evaluation metrics to measure the quality of temporal trajectories from TAROT and other baselines. Specifically, metrics {❶, ❷, ❸} and {❹, ❺} are created to measure the quality of cell lineage and pseudotime, respectively. ❶ *Correlation Test (CT) for Lineages.* We compute the ratio of lineages that pass the correlation test as $\frac{1}{s-1}\sum_t^{s-1}\frac{1}{|\mathcal{T}_{t,t+1}|}\sum_{l_t\in\mathcal{T}_{t,t+1}}\text{CT}(l_t)$, where $\mathcal{T}_{t,t+1}$ is set of mappings $\{l_t:\mathcal{C}_i^{(t)}\to\mathcal{C}_j^{(t+1)}\}$ from time stage $t$ to $t+1$. The $\text{CT}(l_t)$ is the indicator function that returns 1 if the spearman correlation between averaged cell representations from $\mathcal{C}_i^{(t)}$ and $\mathcal{C}_i^{(t+1)}$ is the highest one; returns 0, otherwise. ❷ *Gene Pattern Test per Gene (GPT-G)* and ❸ *Gene Pattern Test per Lineage (GPT-L)*. Based on the developmental and functional priors, we select an extra group of genes for testing, which are not utilized during the TAROT design. The selection follows the widely adopted standards (Finak et al., 2015). Such genes are experimentally validated to have monotonically increased or decreased expressions along with the cell differentiation (or the cell pseudotime). For each test gene, we first compute the percentage of lineages where the gene exhibits monotonicity. Then, averaging the result across all test genes produces the accuracy of GPT-G. Similarly, we first calculate the percentage of genes that exhibit monotonicity along with a given lineage. Then, averaging the result across all inferred lineage generates the accuracy of GPT-L. ❹ *Time Order Consistency Test (TOC) for Lineage.* It examines whether the optimized cell pseudotime is aligned with the time order in lineages. We focus on the tuning point of lineages where the cell differentiation happens *i.e.*, the cell type changes. If the tuning point cluster is $\mathcal{C}_i^{(t)}$, we compute the accuracy of TOC as $\frac{1}{|\mathcal{C}_i^{(t)}|}\sum_{\mathbf{c}_i\in\mathcal{C}_i^{(t)}}\frac{|\{u(\mathcal{G}_i^{(t-1)})<u(\mathbf{c}_i)\}|+|\{u(\mathcal{G}_i^{(t+1)})>u(\mathbf{c}_i)\}|}{|\mathcal{G}_i^{(t-1)}|+|\mathcal{G}_i^{(t+1)}|}$, where $\{u(\mathcal{G}_i^{(t-1)})<u(\mathbf{c}_i)\}$ is a set of cells that belong to $\mathcal{G}_i^{(t-1)}$ and has a smaller pseudotime than $\mathbf{c}_i$. The reported accuracy of TOC is averaged across all tuning points and lineages. ❺ *Temporal Trajectory Error (TTE)* is the average distance between cells to their corresponding projection on the temporal trajectory, *i.e.*, $\sum_{i=0}^{n'}\sqrt{\|\mathbf{c}_i-\mathbb{P}(\mathbf{c}_i,\mathcal{C}(u))\|^2}$. Other details like TAROT's training setups are in Appendix A2.

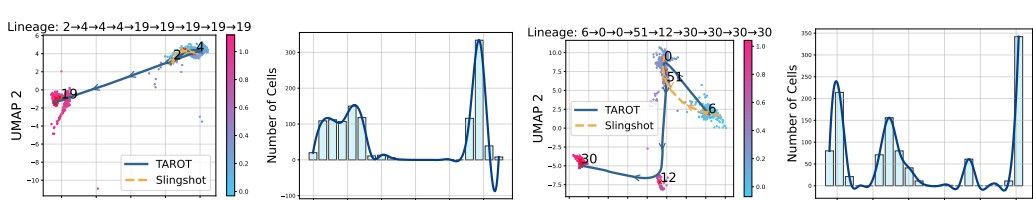

Figure 6: Two inferred lineages and their corresponding pseudotime distributions from TAROT (Ours) and Slingshot (Baseline) on the Mouse-RGC dataset. The color bar indicates the value of cell pseudotime.

### 5.2 SUPERIOR PERFORMANCE OF TAROT IN LINEAGE AND PSEUDOTIME INFERENCE

In this section, we examine the quality of lineage and pseudotime produced by our proposed TAROT. Three representative baselines, *i.e.*, Slingshot (Street et al., 2018), Monocle-3 (Cao et al., 2019), and MOSCOT (Klein et al., 2023), are adopted for throughout comparisons. They are distinctive frameworks based on minimum spanning trees, principal graphs, and optimal transport algorithms, respectively. Experimental results on Mouse-RGC and Mouse-CCC are presented in Figure 6 and Table 1, where several consistent observations can be drawn: ❶ Our TAROT demonstrates great advantages with a clear performance margin compared to Slingshot, Monocle-3 and MOSCOT. In detail, for evaluation metrics {CT (↑ %), GPT-G (↑ %), GPT-L (↑ %), TOC (↑ %), TTE (↓)}, TAROT obtains {65.03%, 19.70%, 18.11%, 20.82%, 1.75}, {63.02%, 13.70%, 11.56%, 44.28%, −0.16} and {3.10%, 16.42%, 16.54%, 37.42%, 0.50} performance improvements on Mouse-RGC and {52.16%, 22, 74%, 11.44%, 25.38%, 0.55}, {27.99%, 35.08%, 27.49%, 34.82%, −0.14} and {9.61%, 22.86%, 6.16%, 30.88%, 0.59} on Mouse-MCC, respectively. Note that a negative TTE gain implies a lower error rate for pseudotime optimization. Such impressive outcomes validate the effectiveness of our proposal. ❷ Although Monocle-3 obtains a lower *Temporal Trajectory Error (e.g.*, 0.18 and 0.14 lower), it fails short in terms of *Time Order Consistency Test* (44.28% and 34.82% worse for the accuracy), compared to our TAROT. It suggests that Monocle-3 probably sac-

Table 1: Performance comparisons of TAROT (Ours) vs. diverse representative baselines on Mouse-RGC and Mouse-CCC datasets. Note that Mouse-iEP is mainly used for a real case study of simulated gene knockout.

| Methods | CT ↑ | GPT-G ↑ | GPT-L ↑ | TOC ↑ | TTE ↓ | CT ↑ | GPT-G ↑ | GPT-L ↑ | TOC ↑ | TTE ↓ |
|---|---|---|---|---|---|---|---|---|---|---|
| | Mouse-RGC | | | | | Mouse-CCC | | | | |
| Slingshot | 5.28 | 41.04 | 42.97 | 72.22 | 2.07 | 10.00 | 67.90 | 77.16 | 68.12 | 1.13 |
| Monocle-3 | 7.29 | 47.04 | 49.52 | 48.76 | **0.12** | 34.17 | 55.56 | 61.11 | 58.68 | **0.44** |
| MOSCOT | 67.21 | 44.32 | 44.54 | 55.62 | 0.78 | 52.55 | 67.78 | 82.44 | 62.62 | 1.17 |
| **TAROT** | **70.31** | **60.74** | **61.08** | **93.04** | 0.28 | **62.16** | **90.64** | **88.60** | **93.50** | 0.58 |

rifices the correctness of pseudotime to better fit the B-Splines. In contrast, TAROT achieves much higher time order consistency with a comparable fitting error, making it a superior choice for neuron trajectory analyses. ❸ Figure 6 presents two examples of inferred lineages and their pseudotime distributions, where TAROT captures a longer range of neuron developmental trajectories.

## 5.3 GENE KNOCKOUT SIMULATION - ALGORITHMIC RECOURSE OF TAROT

With the superior cell lineage and pseudotime from TAROT, we are curious about (1) whether they capture special gene expression patterns; (2) whether these gene patterns are biologically meaningful; (3) how to manipulate them to influence the cell differentiation.

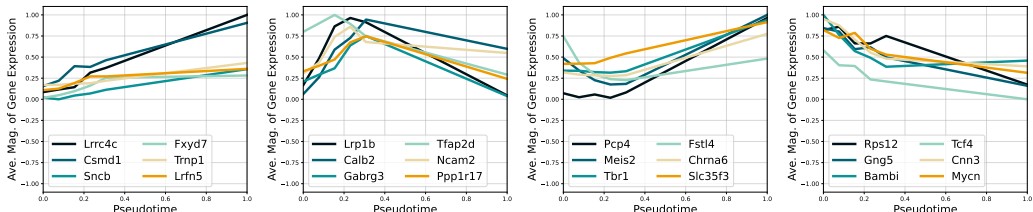

Figure 7: Gene expression dynamics over the cell pseudotime. Four kinds of special gene patterns, from *left* to *right*, are increased, increased then decreased, decreased then increased, and decreased gene waves.

**Gene Pattern Identification.** It is another important angle to dissect the effectiveness of TAROT: examining whether the predicted temporal trajectory can capture clear gene expression patterns. Given one lineage, we record four representative patterns of gene "waves", as presented in Figure 7. We see that in our inferred lineage, the expression values of several gene subgroups consistently increase or decrease, followed by a decrease or vice versa, respectively. For most of TAROT's lineages, a gene set with similar expression patterns can be identified, as shown in Appendix A3. The next step is to validate the biological semantics of these located gene groups.

**Biologically Meaningful? Do the Pathway Alignment.**
We use the GSEA (Fang et al., 2023) for the pathway alignment analysis. It is a method to determine whether the input gene set has statistically significant relationships with pathway gene sets of GO terms in biology. We apply GSEA to the selected gene group from TAROT and Slingshot, and GSEA considers 22 different mouse gene libraries for the alignment. Figure 8 records the top-3 aligned GO terms with the highest gene set overlap ratio. Meanwhile, their normalized p-values are also reported in the $x$-axis. The substantially higher values of both metrics indicate the superiority of TAROT in discovering biologically related gene sets.

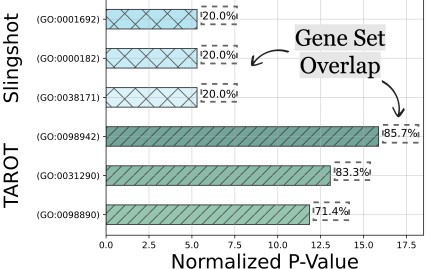

Figure 8: GSEA results of the selected gene sets from Slingshot and TAROT. A higher ratio of gene set overlap and a larger normalized p-value ($|\log_{10}(\text{p-value})|$) indicate a more significant relationship with biologically meaningful GO terms.

**Algorithmic Recourse for TAROT - Simulating the Gene Knockout.** To testify whether the found genes are crucial for the cell temporal trajectory and differentiation, we perform an algorithmic recourse of TAROT by removing these genes during the trajectory optimization to simulate the gene knockout. TAROT results with and without the gene removal are summarized in Figure 9. We can see the trajectory (or differentiation) is significantly altered after even only removing one gene (e.g., TPT1).

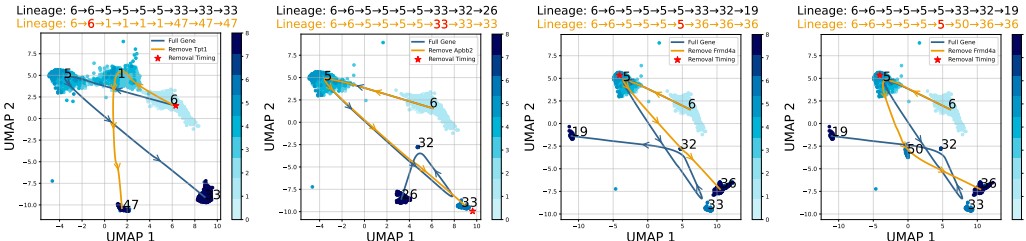

Figure 9: The simulated gene knockout. During the tuning point (red numbers and ⋆) of cell lineage, we remove one of the previously identified genes, which results in totally different cell differentiation.

**A Real Case Study of Gene Knockout.** The next key question is whether our simulated results echo with the wet lab experiment. Rekhtman et al. (1999) provides a real experimental validation on the `Mouse-iPE` dataset (Capellera-Garcia et al., 2016) and proves that knockout genes `GATA1`, `SPI1`, and `LMO2` will discourage the conversion from murine and human fibroblasts to induced erythroid progenitor or precursor cells (iEPs). Impressively, we find that `TAROT` offers aligned simulation results: *removing these genes impedes the mapping (cell differentiation) to the original iEPs*. In details, the initial mappings from `TAROT` are {cell: `Meg` → `Bas`; cell: `Neu` → `Mon`}. If we remove gene `GATA1` and `SPI1`, the simulated results become {cell: `Meg` → `Bas`, `GMP-like`, `MEP-like`; cell: `Neu` → `Mon`, `Bas`, `GMP-like`}. It implies that `TAROT` successfully reveals a seesaw-effect regulation between `SPI1` and `GATA1` in driving the `GMP-like` and `MEP-like` lineages.

## 5.4 ABLATION STUDY

To investigate the contribution of each component in `TAROT`, comprehensive ablations are conducted on `Mouse-RGC`. The studies about different cell representations, biology prior regularization, continuous trajectory optimization, and the automatic thresholding methods in `TAROT` are demonstrated in Table 2, Table 3 and Table A4, respectively.

**Different Cell Representation.** A high-quality cell representation is essential for inferring temporal trajectory. We implement `TAROT` with different representations from diverse sources like `PCA`, `UMAP`, `VAE`, and `MAE`. `PCA` is also applied to `VAE` and `MAE` features to reduce the dimension for a fair comparison. Results in Table 3 evidence the advantage of `MAE`.

Table 2: Ablations on cell representations.

| Mouse-RGC | CT ↑ | GPT-G ↑ | GPT-L ↑ | TOC ↑ | TTE ↓ |
|---|---|---|---|---|---|
| PCA-55 | 69.10 | 53.90 | 54.04 | 74.55 | 0.77 |
| UMAP-2 | 29.44 | 16.16 | 16.14 | 62.69 | 0.58 |
| VAE | 66.43 | 53.02 | 53.07 | 72.17 | 0.85 |
| MAE | **70.31** | **60.74** | **61.08** | **93.04** | **0.28** |

**Different Biology Prior Regularizations.** The flexibility of `TAROT` allows us to add various biology prior knowledge during transport. We investigate different options in Table 3 by adding cost regularizations in an incremental manner. The results tell us that both $\mathcal{D}^{\text{dev}}$ and $\mathcal{D}^{\text{fuc}}$ play significant roles in `TAROT`.

Table 3: Ablations on biology prior regularizations.

| Mouse-RGC | CT ↑ | GPT-G ↑ | GPT-L ↑ | TOC ↑ | TTE ↓ |
|---|---|---|---|---|---|
| $\mathcal{D}$ | 66.28 | 39.60 | 39.60 | 73.41 | 0.64 |
| $\mathcal{D}^{\text{dev}} \odot \mathcal{D}$ | 69.89 | 57.52 | 57.78 | 80.48 | 0.66 |
| $\mathcal{D}^{\text{fuc}} \odot \mathcal{D}$ | 68.71 | 56.01 | 55.76 | 78.62 | 0.72 |
| $(\mathcal{D}^{\text{dev}} + \mathcal{D}^{\text{fuc}}) \odot \mathcal{D}$ | **70.31** | **60.74** | **61.08** | **93.04** | **0.28** |

## 6 CONCLUSIONS

Modeling and inferring single-cell transcriptional patterns is crucial to understanding cell differentiation in developmental biology. This paper presents a novel angle to formulate this fundamental biology problem into a well-defined machine learning formulation - temporal trajectory analysis. We propose a large-scale single-cell dataset of mouse retinal ganglion (`Mouse-RGC`) and an innovative algorithmic framework `TAROT` to: (1) extract superior cell representations; (2) match feature distributions across time stages; (3) optimize and produce continuous temporal trajectories. Extensive investigations validate that our proposals achieve substantial improvements over baseline methods. Lastly, various gene knockout simulations and a real case study are conducted, where the impressive results imply the potential of `TAROT` in providing meaningful biology landscapes. Future work includes more physical validations of mouse gene knockout and potential applications like gene therapy and cell longevity engineering.

## REPRODUCIBILITY STATEMENT

The authors have put in great effort to ensure the reproducibility of algorithms and results presented in the paper. There are detailed explanations of the experiment settings in Section 5.1 and Appendix A2. The study covers 3 distinct datasets, each of which is thoroughly described in Section 3. Furthermore, the evaluation metrics have been clearly introduced in Section 5.1, which serves as a clear framework for evaluating the study's results. Codes are contained in the supplement.

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

## A1 MORE TECHNIQUE DETAILS

**Details about Entropy Weight Search**    The entropy weight $\lambda$ is a critical factor that affects the final Sinkhorn algorithm transport result; an inadequate $\lambda$ makes the transport prone to random mapping. We design a non-linear entropy weight search algorithm to decide an adequate $\lambda$ for the Sinkhorn algorithm. The Pytorch-style pseudo code is presented in Algorithm 1.

---

**Algorithm 1:** Non-Linear Entropy Weight Search in Python style

```python
def nonlinear_entropy_weight_search(lambda, matrix_1, matrix_2, M):
    lambda_i = lambda
    best_cost = infinity
    while True:
        # Calculate sinkhron optimal transport algorithm between matrix_1
            and matrix_2 with cost matrix M with entropy weight lambda_i
        ot_cost, ot_mapping = sinkhorn_optimal_transport(matrix_1,
            matrix_2, M, lambda_i)
        # If not converge, increase entropy weight
        if not converge(ot_mapping):
            lambda_i = lambda_i * 10
        elif ot_cost < best_cost:
            # Decrease entropy weight for better optimal transport mapping
            best_cost, lambda = ot_cost, lambda_i
            lambda_i = lambda_i - lambda_i * 0.1
        # Entropy weight search done
        elif ot_cost > best_cost:
            break
    return lambda
```

---

### A1.1 DETAILS OF THE BASE FUNCTION IN B-SPLINES

B-Spline is constructed based on the base function, and the base function is defined recursively:

$$\mathcal{N}_{i,0}(u) = \begin{cases} 1, u_i \leq u \leq u_{i+1} \\ 0, \text{otherwise} \end{cases} \quad , \tag{3}$$

$$\mathcal{N}_{i,k} = \frac{u - u_i}{u_{i+k} - u_i}\mathcal{N}_{i,k-1}(u) + \frac{u_{i+k+1} - u}{u_{i+k+1} - u_{i+1}}\mathcal{N}_{i+1,k-1}(u), \tag{4}$$

where the $\{u_i\}_{i=0}^{n}$ are the knots of the B-Spline. For more details about adapting the B-Spline for pseudotime trajectory optimization, please check Appendix A2.

## A2 MORE IMPLEMENTATION DETAILS

### A2.1 TRAINING DETAILS OF TAROT

**Pre-Transport - MAE training**    TAROT employs a customized transformer network comprising 6 encoder layers and 6 decoder layers. The encoder layers boast a dimension of 256 with 8 attention heads, while the decoder layer has a hidden dimension of 512 and is also equipped with 8 attention heads. Our MAE uses AdamW (Loshchilov & Hutter, 2019) optimizer with the weight decay of $1e^{-5}$, the learning rate of $1e^{-4}$, and the training step of 50K, wherein the initial 2.5K iterations as a warmup. For the single-cell data, we divide each cell's genes into 128 patches, where each patch contains 64 consecutive gene expression values. The final cell representation $\{\mathbf{c}_i\}_{i=1}^{n}$ is obtained by feeding the encoder output into PCA, which reduces the dimension from 256 to 55.

**During Transport - Regularized OT**    We use Pearson correlation as the vanilla cost function $\text{corr}(\cdot, \cdot)$. The final entropy weight $\lambda$ of each optimal transport is obtained by Algorithm 1.

**Post-Transport - B-Splines Trajectory Optimization**    The curve parameter is predefined before the trajectory optimization. We use 3-degree B-Spline with 300 knots, and the number of learnable control points J is set with 1. The optimization is solved via gradient descent, the learning rate is

$1 \times 10^{-2}$, and the optimization stop condition is the loss fluctuation is less than $1 \times 10^{-4}\%$ with most $1,000$ optimization steps.

## A2.2 DETAILS ABOUT METRIC

For further clarification of differences between GPT-G and GPT-P, we provide the PyTorch-style pseudo codes for both two metrics in Algorithm 2 and 3 respectively.

---

**Algorithm 2:** Gene Pattern Test per Gene in Python style

```python
def gene_monotonically_per_gene(all_lineages, cells, test_gene_group):
    mono_per_gene = []
    for lineage in all_lineages:
        # Calcuate the percentage of genes in test_gene_group, which
            steady increase/decrease over this lineage
        gene_ratio = monotony_in_lineage(lineage, cells, test_gene_group)
        mono_per_gene.append(gene_ratio)
    return mean(mono_per_gene)
```

---

**Algorithm 3:** Gene Pattern Test per Path in Python style

```python
def gene_monotonically_per_path(all_lineages, cells, test_gene_group):
    mono_per_path = []
    for gene in test_gene_group:
        # Calculate the percentage of paths in all_lineages, which the
            specific gene steadily increases/decreases over these paths
        path_ratio = monotonu_of_gene(all_lineage, cells, gene)
        mono_per_path.append(path_ratio)
    return mean(mono_per_path)
```

---

## A3 MORE EXPERIMENTAL RESULTS

### A3.1 MORE RESULTS OF GENE WAVE VISUALIZATION

To illustrate the capability of TAROT in discovering gene sets with specific patterns from lineages, we collect more gene waves with such expression patterns and show them in Figure A11 and A10 in different forms. Results show that TAROT yields more genes with similar expression patterns since the high-quality lineage and pseudotime inference.

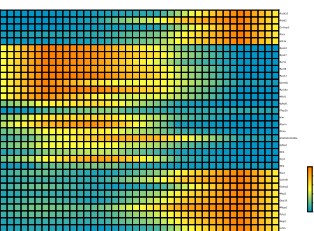

Figure A10: Gene expression dynamics over the cell pseudotime *left* to *right*. From *top* to *bottom* are the heatmap of different genes that increase, decrease, increase followed decrease, and decrease followed increase.

### A3.2 MORE ABLATION RESULTS

**Different Options for Pseudotime Trajectory Optimization** The number of learnable points J between two fixed control points greatly affects the B-Spline pseudotime trajectory optimization. We ablate different J to seek a plausible setting for the pseudotime trajectory optimization. Table A4 indicates that more learnable control points deliver improved TTE but sacrifice TOC, which indicates better trajectory fitting does not result in better pseudotime trajectory. Therefore, we use "1" learnable control points per two fixed points. We also compare our method with two other trajectory fitting methods: the "Poly." method, which uses the polynomial curve for temporal trajectory fitting (the degree of curve is the number of cell differentiations that happen), and the "Principal" method, which uses the principal curves algorithm, the same trajectory fitting method with Slingshot (Street et al., 2018).

Table A4: Result of different trajectory optimization options.

| Methods | TOC ↑ | TTE ↓ |
|---------|-------|-------|
| | **Mouse-RGC** | |
| Sp-1 | **93.04** | 0.28 |
| Sp-2 | 90.73 | 0.25 |
| Sp-3 | 92.03 | **0.23** |
| Poly. | 63.37 | 1.24 |
| Principal | 72.22 | 2.07 |

**Automatic thresholding within TAROT** In TAROT, the automatic thresholding method is vital to achieving accurate lineage results. We proposed two candidate automatic thresholding techniques:

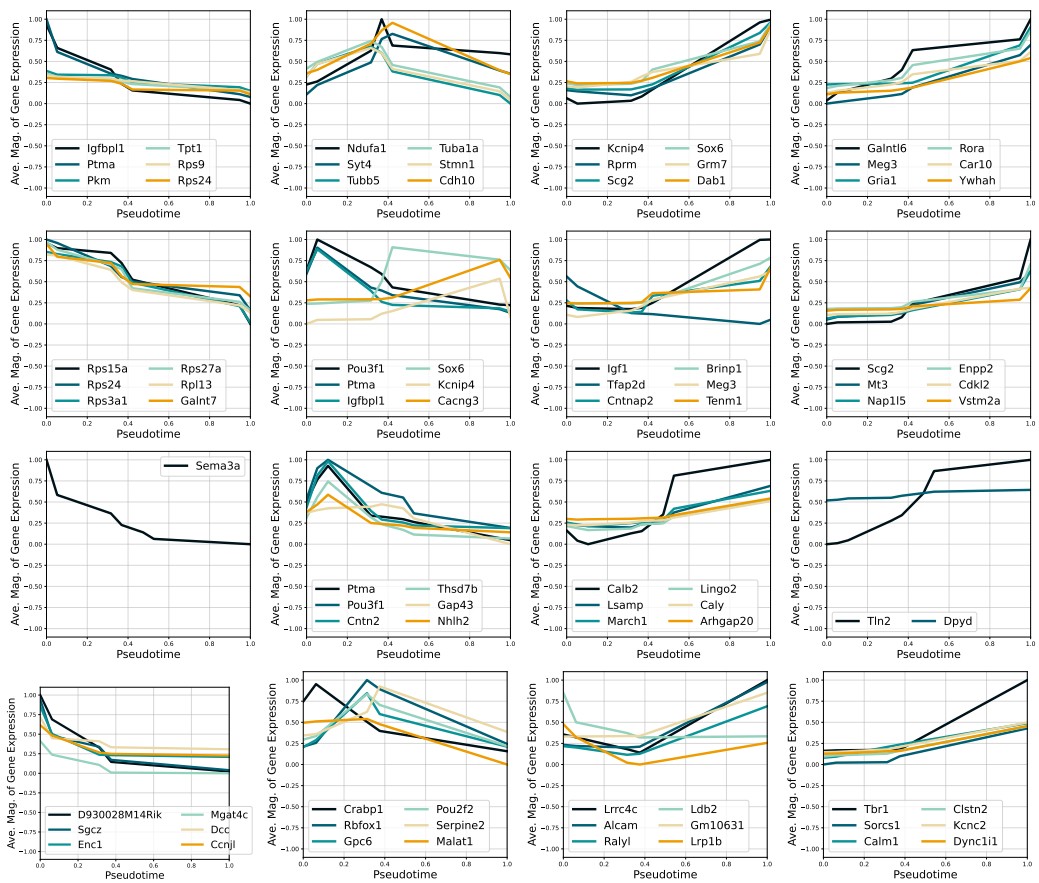

Figure A11: More gene expression dynamics over the cell pseudotime. Four kinds of special gene patterns, from *left* to *right*, are increased, increased then decreased, decreased then increased, and decreased gene waves. Gene waves in different lines are identified from different lineages.

the "P-value" method and the "max. sep." method (*i.e.*, the maximum separation). The "P-value" method utilizes statistical significance to identify mappings with a p-value lower than the threshold of $1e^{-4}$. Conversely, the "max. sep." method selects mappings with close OT costs but notably distinct from other mappings, emulating human intuition. Table A4 reports TAROT's results with two thresholding methods, demonstrating that the "max. sep." selects lineages with higher quality.

## A4  ETHICAL STATEMENT ABOUT DATASET COLLECTION

For the data collection of `Mouse-RGC`, mice were maintained in pathogen-free facilities under a 12-hour light-dark schedule with standard housing conditions. Food and water were continuously supplied. Animals used in this study include both males and females.

Table A5: Ablations on automatic thresholding.

| **Mouse−RGC** | CT ↑ | GPT-G ↑ | GPT-L ↑ | TOC ↑ | TTE ↓ |
|---|---|---|---|---|---|
| P-value | 66.58 | 56.30 | 56.89 | 92.86 | **0.23** |
| max. sep. | **70.31** | **60.74** | **61.08** | **93.41** | 0.30 |

