# OpenReview forum: "Regularized Optimal Transport for Temporal Trajectory Analysis in Single-Cell Data"
_ICLR.cc/2024/Conference — Submitted to ICLR 2024_

### Official Review · Reviewer_nX3H · 2023-10-27

**Soundness:** 2 fair
**Presentation:** 3 good
**Contribution:** 2 fair
**Rating:** 3
**Confidence:** 3

**Summary:**

In this manuscript the authors (1) present a new dataset (Mouse-RGC) for benchmarking single-cell trajectory inference methods and (2) propose TAROT, a framework for performing trajectory inference. The authors apply TAROT to two real-world datasets and compare their method's results with those of moscot (another optimal-transport-based framework) and Slingshot/Monocle3 (two non-OT based frameworks).

**Strengths:**

* **Clarity:** The manuscript was very well-organized and easy to follow. Well done!
* **Significance:** The problem studied by the authors (i.e., inferring developmental trajectories) is of major importance in single-cell biology.
* **New dataset contribution:** As part of their work, the authors propose a new benchmarking dataset which will likely be of great interest to the single-cell community. (I did not see this explicitly stated in the manuscript, but I am assuming that the dataset will be made publicly available.)
* **Ablations**: I appreciated the ablation studies for better understanding how the individual components of the proposed method affect the overall performance.

**Weaknesses:**

Despite the strengths of the paper, I believe that its flaws outweigh its strengths at this time. In particular, I believe there are nontrivial issues with the experimental validation, as well as the claims made in terms of methods development. Details below:

* **Claims re: methological novelty:** From my reading of the manuscript, the computational novelty in TAROT consists of (1) using a masked autoencoder for learning representations of cells, and (2) incorporating the "Developmental" and "Gene expression" cost function regularizers. On the other hand, casting trajectory inference as an optimal transport problem is certainly not novel (see e.g. [1, 2]) and using splines to assemble continuous trajectories has also been done in previous work (e.g. [3]); however, the wording throughout the paper seems to imply that these ideas are also novel (see e.g. the summary of contributions at the end of Section 1). I believe it's important that the authors clarify _exactly_ what their contributions are in terms of methods development here.
* **Weaknesses in quantitative evaluation metrics:** The metrics used for evaluation in Table 1 seem a bit unfair towards other methods. For example, The GPT-G and GPT-L metrics are essentially evaluating whether each method's results agrees with the prior knowledge that's explicitly built into the TAROT cost function. In other words, the metrics assume that this prior knowledge can be used as a gold-standard ground truth for evaluating trajectories, which isn't so clear-cut to me. For me to be convinced of TAROT's superiority over previous methods, I believe the authors would need to evaluate their methods based on more solid ground truths (e.g. from simulated data, or known developmental trajectories in real-world data as in Fig. 2 of [2]).
* **Weaknesses in qualitative evaluations:** While appreciate the goal of the authors qualitative evaluations in Section 5.3, I believe again there are issues here that prevent me from concluding that TAROT is superior to previous methods. For example, in the gene pattern identification in Figure 7, the authors note that TAROT captures distinct "wave" patterns of gene expression varying over pseudotime (including _non-monotonic patterns_), and conclude that this phenomenon validates TAROT's efficacy. Based on the authors assumption that gene expression should vary _monotonically_ over time, framing this finding as a positive thing seems odd and potentially suggests issues with the metrics noted in my previous point. Other issues include: not including Monocle-3/moscot in the GSEA comparison of Figure 8, not discussing whether the specific pathways found in the GSEA analysis are meaningful/align with biology, and not comparing with any other methods in simulated gene knockout experiments.
* **Issues with the constructing clusters for the M-RGC dataset:** The construction of the "cell type" clusters for the M-RGC dataset is a bit arbitrary, and not entirely grounded in known biology. For example, the authors use $k=50$ neighbors for constructing a nearest-neighbor graph. This choice of $k$ is justified according to the "experience of biological sciences", though popular packages like Seurat use other values (e.g. 20 in https://satijalab.org/seurat/reference/findneighbors). Similarly, Louvain results are highly sensitive to the choice of resolution parameter, and I could not find discussion anywhere of how this parameter was chosen. Ideally, these clusters should be chosen/validated using known marker genes before using them as ground truths.

[1]: Schiebinger et al. "Optimal-Transport Analysis of Single-Cell Gene Expression Identifies Developmental Trajectories in Reprogramming"
[2]: Klein et al. "Mapping cells through time and space with moscot"
[3] Trapnell et al. "The dynamics and regulators of cell fate decisions are revealed by pseudotemporal ordering of single cells"


Minor points:
* **Missing dataset?** The authors mention in the summary of contributions in the introduction that they validate TAROT on the Mouse-RGC dataset and two publicly available datasets, but I only saw one publicly available dataset in the results section (Mouse-CCC).
* **MCC vs. CCC?** Related to the previous point, the authors alternate between referring to a Mouse-MCC dataset vs. Mouse-CCC dataset, but I only see results for Mouse-CCC in Table 1.

**Questions:**

* Could the authors clarify in more detail how exactly their method differs from previous computational works for inferring cellular trajectories?
* Did the authors validate the clusters in the M-RGC dataset with known biological phenomena?
* Are results from Mouse-MCC/CCC missing?

---

### Official Review · Reviewer_zx3P · 2023-10-28

**Soundness:** 2 fair
**Presentation:** 1 poor
**Contribution:** 2 fair
**Rating:** 3
**Confidence:** 3

**Summary:**

The paper presents $\rm{TAROT}$, a framework to model and infer the evolution trajectory of single-cell data.  $\rm{TAROT}$ consists of (i) learning cell representations using a Masked Autoencoder Transformers, (ii) applying regularized optimal transport which incorporates biological priors, and (iii) continuous trajectory optimization via B-splines. The framework is evaluated over a novel dataset,  $\rm{Mouse-RGC}$, a large-scale mouse retinal ganglion cell dataset with annotations for nine time stages and 30, 000 gene expressions as well as two publicly available single-cell datasets.

**Strengths:**

$\underline{\textrm{Originality}}$: the paper’s originality comes from suggesting a unifying framework to model and infer the evolution trajectory of single-cell data which combines three tasks that are often considered independently. This approach enables the optimization of each step with respect to the subsequent task.

$\underline{\textrm{Quality}}$: the paper presents a high-quality analysis. First, it examines multiple datasets and baseline methods, evaluating them using diverse metrics to quantitatively assess various aspects of temporal trajectory quality. Additionally, biological aspects of learned trajectories are studied and framework ablation studies are provided.

$\underline{\textrm{Clarity}}$: the framework components are presented in a clear manner.

$\underline{\textrm{Significance}}$: the significance of the paper parallels its originality, as it introduces a unifying framework that combines three crucial components: cell-representation construction, optimal transport based mapping across time points, and continuous trajectory inference using b-splines interpolation.

**Weaknesses:**

1. The different components of the $\rm{TAROT}$ algorithm: while there is a benefit in considering the complete pipeline, as suggested in $\rm{TAROT}$ it is important to acknowledge existing community efforts. See following points.
Representation learning in single-cell genomics: as outlined by the authors (Section 2) much effort has been put to improve the quality of learned cell representation. However, the authors do not utilize an existing tool (which is already optimized for sc-data and usually allows incorporation of prior biological knowledge) nor do they consider any of the existing tools in the presented ablation study (Section 5.4). This limits the ability to judge the optimality of the claim of MAE being a “Superior cell representation”.
2. OT application: Similarly to the previous point, as mentioned by the authors application of OT for trajectory inference in single-cell data was already suggested in WOT (Schiebinger et al. 2019) and MOSCOT (Klein et al., 2023). Importantly, all settings consider regularized optimal transport to perform the matching and the difference boils down to the biological priors used to regularize the cost. In $\rm{TAROT}$  this prior is detailed in the section “Biology Priors Regularize Cost Function”. However, the option to introduce a similar prior in previous frameworks is not mentioned. It is important to note that MOSCOT allows for the computation of OT marginals from proliferation and apoptosis markers in developmental processes to capture the different proliferation and death rates of cells. It will be valuable to add a description of previous methods, stand on the main difference between them and the current approach, and elaborate on the setting chosen to evaluate the baseline, MOSCOT.
3. Biological priors: the authors define two biological priors used to regularize the cost matrix (1) Developmental and (2) Gene expression. As these stand out as the major contribution of the framework it will be beneficial to provide further details on the construction of these regularizers, e.g. how is the regularization introduced in practice and how was the gene-list constructed.
4. The entropy regularization parameter ($\lambda$): the authors acknowledge the importance of the entropy regularization parameter and present a “Non-Linear Entropy Weight Search” procedure used to set it (Algorithm 1). With that, it will be valuable to present ablation with respect to this value, given its importance in the OT setting, or at least report the value used in presented results.
5. Data and code availability: it will be beneficial to accompany the attached code with the relevant data files. Specifically the mouse-RGC dataset could be a useful resource for the community.
6. Writing quality: the paper contains many grammatical errors, typos, and misuse of terminology. Hence,  there is a need for comprehensive proofreading and language refinement.

**Questions:**

In light of the above weaknesses, beyond an overall language refinement, the following will be valuable to improve the paper quality:
1. Additional ablations with respect to the cell-representation.
2. A dedicated section discussing the baseline methods.
3. An elaboration on the biological priors used for the OT cost matrix.
4. Relating to $\lambda$ values used in practice.
5. Sharing the mouse-RGC dataset.

---

### Official Review · Reviewer_waBn · 2023-10-31

**Soundness:** 3 good
**Presentation:** 3 good
**Contribution:** 3 good
**Rating:** 5
**Confidence:** 3

**Summary:**

This paper studies the understanding of the temporal relationship between cellular states and lineages.  The authors constructed a large-scale mouse retinal ganglion cell dataset for research. Additionally, an optimal transport-based algorithm, TAROT is presented, in which a masked autoencoder is devised to learn representations, biology prior-based regularization on cost function is designed, and B-Splines-based continuous trajectory optimization is presented. I am happy to see the contribution of the authors to the community. However, I have some concerns (see weakness).

**Strengths:**

* A large-scale dataset is constructed.
* The proposed OT algorithm is effective as shown in experiments.
* The paper is well-written and easy to follow.

**Weaknesses:**

I suggest the authors to release the dataset. Besides, I have the following concerns:
* The motivation of the MAE could be more clear. The difference and relation to "Masked Autoencoders Are Scalable Vision Learners (CVPR 2022)" should be discussed, for which I found no citation.
* Why are the MAE representations better than the autoencoder or PCA for the gene data? MAE is effective for vision data. Could the authors provide some comments on the effectiveness of MAE for gene data?
*  Is $\mathcal{C}\_1^{(t)},...,\mathcal{C}\_{k_t}^{(t)}$ clustered out for each time stage $t$ individually rather than clustering all the time-stage cells?
* Since the OT is developed on the clusters, the results could highly rely on the quality/correctness of the clustering. Meanwhile, I could not find the clustering details.
* The details of $\mathcal{D}^{\mbox{\tt dev}}$ and $\mathcal{D}^{\mbox{\tt fuc}}$  is not given. Besides, the expression of $\mbox{\tt corr}()$ is missing.
* The motivation of B-Splines is not clear. Meanwhile, I do not find how the transport plan $\mathcal{T}$ is applied in the B-Spline.
* How to understand the "pseudo time".

I am happy to increase the score if the concerns are adequately addressed.

**Questions:**

See weakness.

---

### Meta-Review · Area_Chair_7HMk · 2023-12-12

**Metareview:**

The paper proposes an application of OT algorithms to reconstruct lineage of single-cell data. Reviewers point out a few weaknesses (lack of motivation at some steps, comparison with existing baselines or reporting of hyperparameters) which led them to a fairly low evaluation, as well as comments that could be useful for a second round. The authors did not provide a rebuttal.

**Justification For Why Not Higher Score:**

absence of rebuttal prevented any score increase

**Justification For Why Not Lower Score:**

NA

---

### Decision · Program_Chairs · 2024-01-16

Reject